# An Alternative Digital Image Correlation-Based Experimental Approach to Estimate Fracture Parameters in Fibrous Soft Materials

**DOI:** 10.3390/ma15072413

**Published:** 2022-03-25

**Authors:** João Filho, José Xavier, Luiz Nunes

**Affiliations:** 1Laboratory of Opto-Mechanics (LOM), Department of Mechanical Engineering (PGMEC-TEM), Universidade Federal Fluminense (UFF), Rua Passo da Pátria, 156, Bloco E, Sala 210, Rio de Janeiro 24210-240, Brazil; luizcsn@id.uff.br; 2Research and Development Unit for Mechanical and Industrial Engineering (UNIDEMI), Department of Mechanical and Industrial Engineering, NOVA School of Science and Technology, NOVA University Lisbon, 2825-149 Lisbon, Portugal

**Keywords:** mode I fracture, crack tip opening displacement, crack tip opening angle, digital image correlation, fibrous soft composite

## Abstract

One of the main challenges in experimental fracture mechanics is to correctly estimate fracture parameters of a nonhomogeneous and nonlinear material under large deformation. The crack tip detection is strongly affected by fibers at crack tip, leading to inaccurate measures. To overcome this limitation, a novel methodology based on the Digital Image Correlation (DIC) method for crack tip detection of fibrous soft composites is proposed in this work. The unidirectional composite was manufactured using a matrix of polydimethylsiloxane reinforced with a single layer of extensible cotton knit fabric. For two different fiber orientations, the crack growth (da), Crack Tip Opening Displacement (CTOD) and Crack Tip Opening Angle (CTOA) were determined using pure shear specimens under mode I fracture. A consistent estimation of fracture parameters was obtained. The location of the crack tip position during the fracture test using the DIC-based methodology was validated against a visual inspection approach. Results indicated that the DIC-based methodology is easily replicable, precise and robust.

## 1. Introduction

Over the past years, a great effort has been made to investigate the mechanical behaviour of biological tissues, towards a better description of risk of rupture [1]. In particular, there is a growing interest in studying the fracture behaviour of soft tissues for a better understanding, modelling and replacement. For instance, the progressive damage propagation of diseased human thoracic aortic fibrous tissue [2], the measurement of fracture properties and dissection of arteries [3], strategies to prevent the tearing of skin [4], the tear resistance of tendons and ligaments [5], the rupture of membranes [6] and the laceration of perineal tissue [7] were recently investigated. It is known that such materials exhibit a complex non-linear behaviour in which the classical linear fracture mechanics cannot be considered [8].

Several approaches have been already reported in the literature to address the measurements of important fracture parameters from displacement fields provided by Digital Image Correlation (DIC) [9,10]. Among them, the Crack Tip Opening Displacement (CTOD) has been estimated from displacement full-fields during a fracture test using this non-contact method [11]. This methodology contrasts with more conventional techniques, in which CTOD is determined by crack opening clip gages, considering a linear behavior. The assessment of CTOD has allowed a direct approach for determination of the cohesive laws under different failure models, i.e., mode I [12], mode II [13], and mixed mode I + II [14]. Moreover, the evaluation of the crack path variation during the fracture test in mode I [15] and mixed mode I + II [16] have been achieved. However, fibrous soft biological tissues and synthetic soft composite materials exhibit a complex nonlinear behavior. Therefore, important fracture parameters such as the CTOD, Crack Tip Opening Angle (CTOA) and crack growth (da) are difficult to be extracted in this type of material. The correct estimation of these parameters plays an important role in understanding and modelling such complex systems. Although several estimation procedures have been proposed in the literature [17] for general purposes, they are expected to fail in the presence of fibrous soft materials. This problem arises because fibrous soft materials can undergo large deformations and fibers are the principal source of anisotropy. The lack of information regarding the estimation of such parameters taking into account fiber-reinforced materials highlights the important gap in non-linear fracture mechanics that should be further investigated.

The crack tip detection is extremely difficult to be computed in experiments [18]. There are several methodologies that can be used to extract the crack tip position and propagation, assisted by camera-, infrared-, ultrasonic- and laser-based techniques [17]. Among them, automatic image-based algorithms have been recently proposed and developed to overcome traditional approaches. One of the main advantages of using image-based approaches is the accurate detection of the crack tip. The *ImageJ* open-source software is often used to perform image segmentation to isolate the crack [19,20]. For a further analysis, the DIC technique has been used to measure the crack propagation from full-field displacements with sub-pixel resolution [9,21]. The detection can be carried out using the concept of the maximum slopes of the displacement oriented at the notch direction [22]. The quantitative evaluation of the crack position can be encountered using a mask field with a predefined constant threshold value using DIC [15]. Moreover, the DIC method has been employed to measure the crack growth of a soft homogeneous silicone elastomer using an iterative numerical algorithm [23]. Besides, the crack tip opening displacement has been computed from DIC measurements to directly determine cohesive laws of biological tissues [12].

For nonhomogeneous materials, the presence of fibers at the crack tip can lead to a challenging crack tip detection. The commonly used techniques in literature, such as image segmentation, cannot be implemented in these fiber-reinforced materials. Therefore, the main purpose of this work is to propose an alternative DIC-based methodology for crack tip detection of soft composites under mode I fracture. Due to difficulties on handling and testing biological samples [24], synthetic silicone-based elastomer was proposed to mimic their hyperelastic behaviour [25]. Since this rubber-like material was used to compose the matrix of the considered soft composite, the fibers are responsible to introduce nonhomogeneity and anisotropy. The crack growth (da), CTOD and CTOA were estimated from the crack tip detection.

## 2. Materials and Methods

### 2.1. Specimen Fabrication and Experimental Setup

The soft composite was manufactured from a matrix of polydimethylsiloxane (silicone rubber model 4–150 RTV from *Moldflex*, São Paulo–Brazil) reinforced with a single layer of extensible cotton knit fabric. Figure 1a illustrates the structure and density of the cotton fabric. A fiber volume fraction on the order of 4% was obtained. This composite was selected since its behaviour can be approximated to a fibrous soft biological tissue, which is composed of a cellular matrix reinforced by collagen fibers [26]. In literature, fibrous soft biological tissues are often treated as fiber-reinforced incompressible rubber-like materials [24]. The liquid silicone rubber was mixed with catalyst considering a mass ratio of 100:3. A flat piece of glass coated with mold release wax (model *TecGlaze-N* from *Polinox*, São Paulo–Brazil) was used to hold the soft composite. The manufacture process was carried out in 3 main steps: first, (1) a single layer of silicone mixture was equally distributed on the glass; then, (2) the cotton knit fabric was gently placed on the matrix using a resin roller and (3) the mixture was deposited above the two layers to seal up the fibers. The composite was cured at room temperature for 48 h before unmolding process. Although the cotton fabric was made of unidirectional fibers, in the perpendicular direction there were small random filament sets responsible to maintain the arrangement of the warp knitted structure, as depicted in Figure 1a. Therefore, two fiber orientations (θ) were considered: $0^{\circ }$ (fiber direction) and $90^{\circ }$ (transverse direction). Four planar specimens with nominal dimensions of 100 × 60 × 0.8 $\;{\mathrm{mm}}^{3}$ for each fiber orientation were fabricated. Figure 1b illustrates the soft composite.

Several test and geometrical configurations have been proposed in the literature for both pure and mixed-mode fracture loading conditions [27,28]. For mode I fracture loading, a comparison between pure shear (PS) and single edge notch tension (SENT) specimens using rubber-like materials was recently performed [29]. PS specimens presented advantages over SENT specimens in view of the fracture characterization of materials. In general, accurate measurement of the crack growth is more difficult to be reached using SENT than PS specimens. For this reason, the pure shear (PS) configuration (see Figure 2a) was chosen in the present work. The effective area (area between clamps) of 100 × 10 $\;{\mathrm{mm}}^{2}$ was considered and a small pre-crack ($a_{0}$) of 20 mm was made in specimens using a sharp blade to induce mode I fracture. No damage was seen in the crack interface during the cutting process. The initial width, denoted by *L*, must be larger than the initial effective height ($h_{0}$) in order to guarantee a pure shear state on the planar specimens [30].

The fracture tests were conducted at room temperature ($\approx 25^{\circ }$) under quasi-static loading condition using a constant cross-head velocity of 4 mm/min. Figure 2b shows the experimental arrangement, coupling the fracture test with the DIC optical system. A monochrome high-resolution camera (model Basler Ace model acA1300-200 µm) with the output resolution of 1280 × 1024 pixels^2^ and cell size of 4.85 × 4.85 $µm^{2}$ was used together with a 1/2″ 13–130 mm 10X Close-up Manual Zoom lens (model MLH-10X from Computer). The axis of the camera-lens set was positioned perpendicular to the plane surface of the PS specimen and the image acquisition was configured to capture the first 20 images every 5 s and the remaining every 1 s until complete failure. To reduce the effect of lens distortion on the region of interest, the camera was placed facing the initial crack tip ($a_{0}$). A 500 N load cell was used to register the applied load simultaneously. Red light sources was chosen to reduce undesirable fluctuation of light and improve the quality of images (Figure 2b), knowing that the camera sensor is more sensitive to the red wavelength (≈660 nm). Images were recorded and processed using a DIC algorithm implemented in python language, so-called *iCorrVision-2D*. This software is open-source and developed on a subset-based DIC approach.

Since a planar specimen was used, the 2D-DIC method can be employed in this work. DIC is a non contact optical method capable of measuring full-field displacements and strains of a specimen being tested [21]. The crack growth, commonly denoted by da, the CTOD and the CTOA fracture parameters can be computed by means of DIC. It is important to remark that, considering soft materials, the use of classical local measurement techniques are not recommended or even possible. Besides the fixture complexity, large displacements can be hard to be extracted using these techniques. All PS specimens were coated with an overspray of black paint to produce a random speckle pattern in view of improving the matching between images. Figure 1c illustrates the produced speckles with average size of 4.3 pixels and black/white ratio of approximately 1.04. The *iCorrVision-2D* DIC software was used to compute the in-plane measurements from the captured digital images. Table 1 shows the adopted DIC settings to configure the software. It should be highlighted that this work followed the guidelines proposed by the International Digital Image Correlation Society (iDICs) for DIC-based measurements [31]. The incremental approach, in which the correlation process was performed on subsequent image pairs, i.e., $i_{n}$ and $i_{n+1}$ (*n* is the instant of time), was used instead of spatial correlation (fixed undeformed image $i_{0}$ as reference) since large displacements can lead to a significant loss of correlation [32].

### 2.2. Evaluation of Crack Growth (da), CTOD and CTOA

The DIC-based crack tip detection can be carried out using virtual extensometers. Figure 3 illustrates this idea by showing the linear spatial distribution of extensometers, defined in the image between DIC data points (represented by a red cross symbol) selected at the top and bottom of the visual crack path. At these points, the local correlation is performed by means of the DIC algorithm and the displacement (u) of points can be computed according to the following expression,
(1)$$\begin{array}{c} u(X,t)=x(X,t)-X \end{array}$$
where x is the current (deformed) configuration, X is the reference (undeformed) configuration and *t* is the incremental time. From the reference and current positions of the DIC calculation points, the Euclidean distance between each pair of points can be measured and the vertical displacement (VD) can be easily extracted, according to,
(2)$$\begin{array}{c} \mathrm{VD}\left(k,i_{n}\right)={\left[{\left(x_{11bk}-x_{11tk}\right)}^{2}+{\left(x_{22bk}-x_{22tk}\right)}^{2}\right]}_{i_{n}}^{\frac{1}{2}}-\mathrm{VD}\left(k,i_{0}\right) \end{array}$$
where subscripts *t* and *b* are related to the DIC data points at the top and bottom of the crack path, respectively, *k* is the DIC point index, $i_{n}$ is the image captured at instant *n*, $\mathrm{VD}(k,i_{0})$ is the initial Euclidean distance between the reference top and bottom DIC subset calculation points obtained from image $i_{0}$, and $x_{11}$ and $x_{22}$ are their coordinates in the image plane. Therefore, VD can be defined as a virtual displacement gauge (VDG) positioned along the crack path (see Figure 3). It should be mentioned that the vertical displacement in the crack region is often expressed in literature as the crack opening displacement (COD) (Appendix A).

From the calculated VD curves, the first step towards the detection of the crack tip consists of taking the mean value of VD (denoted here by $\bar{\mathrm{VD}}$) as follows,
(3)$$\begin{array}{c} \bar{\mathrm{VD}}\left(i_{n}\right)=\frac{1}{k}\sum_{k=1}^{k}\mathrm{VD}\left(k,i_{n}\right). \end{array}$$

The $\bar{\mathrm{VD}}$ can then be used as a threshold value to estimate the crack tip from the VD curves at each instant *n* as function of $x_{11}$ (crack path direction) [15,18]. However, due to the complex nonlinear behaviour of fiber reinforced soft composites, the threshold value need to be adjusted, in this case, using a linear function. Therefore, the adjusted threshold line-cut (${\mathrm{VD}}_{th}$) is given by,
(4)$$\begin{array}{c} {\mathrm{VD}}_{th}\left(i_{n}\right)=\bar{\mathrm{VD}}\left(i_{n}\right)\left[\alpha i_{n}+\beta \right] \end{array}$$
where α and β are the respective angular and linear coefficients defined by,
(5)$$\begin{array}{c} \alpha =\frac{w_{0}-w_{f}}{i_{0}-i_{f}}\;\mathrm{and}\;\beta =w_{0}-\alpha i_{0} \end{array}$$
where $w_{0}$ and $w_{f}$ can be expressed as a factor used to adjust the model to the initial ($p_{0}$) and final ($p_{f}$) crack tip positions, respectively, along $x_{11}$ axis. A virtual manual detection is performed to initialize $p_{0}$ and $p_{f}$. Figure 4a depicts the construction of the proposed linear function. The factors $w_{0}$ and $w_{f}$ must be calibrated for each specimen.

Figure 4b graphically illustrates some details of the proposed methodology for the crack tip detection using the VDG approach under mode I fracture. The intersection between each ${\mathrm{VD}}_{th}\left(i_{n}\right)$ line-cut and the correspondent $\mathrm{VD}(k,i_{n})$ curve at instant *n* represents the $x_{11}$ position of the crack tip, denoted by ($p_{n}$).

Figure 5 illustrates the three fracture parameters: da, CTOD and CTOA. The crack growth at instant *n* can be obtained according to,
(6)$$\begin{array}{c} da_{n}=p_{n}-p_{0}. \end{array}$$

The CTOD can be determined from a pair of DIC calculation points near the initial crack tip ($p_{0}$) and CTOA can be estimated from the angle formed between the tangent lines to the DIC calculation points in which the CTOD was extracted (Figure 5). Here, CTOD is equal to $\mathrm{VD}(k_{p_{0}},i_{n})$ calculated at each instant *n* considering the position of initial crack tip ($p_{0}$) and CTOA extraction is based on the curvature of the crack opening shape. Commonly, analytical expressions are implemented in literature to compute CTOA. However, these expressions cannot be used in the soft composite studied in this work. The complex crack growth and the presence of stretched fibers in the crack interface can compromise the correct extraction of analytical CTOA.

## 3. Results and Discussion

The anisotropic hyperelastic material used in this work exhibited a complex response when subjected to mode I loading. For illustrative purposes, Figure 6 and Figure 7 depict the respective crack opening shape of the soft composite with fiber oriented at $0^{\circ }$ and $90^{\circ }$ with respect to the direction of applied load for different loads. As can be noticed, the crack opening shape and crack growth are dependent on the fiber orientation.

For both specimen configurations, a blunt crack can be observed at the onset of crack propagation. However, after large crack growth that can be seen on specimens with fiber orientation of $90^{\circ }$ (see Figure 7), the presence of fibers induces the propagation of a typical sharp crack. Therefore, the crack is expected to propagate at lower loads considering the fiber orientation of $90^{\circ }$. This complex behaviour makes it difficult to correctly estimate the position of the crack tip, as well as all fracture parameters.

Due to the presence of stretched fibers in the crack boundaries, classical image-based methodologies for crack tip detection are expected to fail. For instance, image segmentation using threshold functions cannot remove the stretched fibers from the captured images and loss of resolution of the crack boundaries can be observed. Figure 8a,b show images of the soft composite with fibers oriented at $0^{\circ }$ and $90^{\circ }$, respectively, where the red cross symbol represents the crack tip position obtained by manual inspection. The results of image segmentation are illustrated in Figure 8c,d. The image procedures were performed using *ImageJ* and Python. It is clear that the random fiber distribution along the crack opening region creates a random pattern that can compromise the image correlation. As can be seen, the crack tip position is underestimated after image segmentation using DIC (blue cross in Figure 8c,d). Usually, image segmentation-based techniques can be implemented with homogeneous materials. However, the use of these techniques regarding fibrous soft composites should be avoided.

Figure 9a,b illustrate the crack growth (da) as function of the number of captured images ($i_{n}$) for fiber orientations of $0^{\circ }$ and $90^{\circ }$, respectively. These results were obtained from the proposed DIC-based crack tip detection methodology using a linear correction function. The continuous curve depicts the mean value and standard deviations of the estimated da. The values of da were also evaluated from the crack tip positions, which were manually detected for each specimen (SP0 for $0^{\circ }$ and SP90 for $90^{\circ }$). Here, manual detection of some points was carried out only for verification and discussion purposes. As can be observed, the proposed method can detect the crack tip position with consistency and precision. Apart from SP0 3 (shown in Figure 9a), all scattered data are included in the standard deviations. It should be highlighted that the high standard deviations obtained by the proposed DIC-based approach followed the dispersion of the manually detected crack tip.

From the extraction of the initial crack tip position, CTOD can be easily computed using VD. However, it is important to firstly investigate the most appropriate region for positioning the DIC calculation points. Therefore, 10 points oriented at the $x_{22}$-direction were selected at the top and bottom regions of the initial crack tip. The Euclidean distance of each pair of points from the outer boundaries to the crack tip is depicted in Figure 10a. As can be seen, the calculated CTOD from these pairs of points are close to each other. Clearly, from the mean value of CTOD, the middle set of points (blue) delimits the most appropriate region to construct the DIC calculation points. Moreover, the points near to the crack tip (green) should be avoided since the crack propagation can induce a loss of correlation in this region. Figure 10b illustrates the typical CTOD-da curve for fiber oriented at $0^{\circ }$ and $90^{\circ }$. Up to 1 mm, both configurations exhibit the same behaviour, which start to deviate afterwords since the crack growth resistance of soft composite depends on fiber orientation.

CTOA can be straightforward computed by constructing lines tangent to the CTOD calculation points and then extracting the angle between both lines (see Figure 5). Figure 11a depicts the mean CTOA value and standard deviation as function of the crack growth for both fiber orientations using the proposed approach. Figure 11b shows the mean CTOA as a function of the mean CTOD, together with standard deviation. It was noticed in experiments that after reaching the maximum CTOA value, the crack shape changed from blunt to typical sharp crack (see Figure 6 and Figure 7). The observed shape change produced a maximum point in the curves depicted in Figure 11. As can be seen, this particularity became more evident for fiber oriented at $90^{\circ }$. In fact, the geometrical DIC-based approach seems to be an appropriate method to use in fibrous soft composites. It can be observed that the results have an interesting repeatability that can be hard to be noticed in soft composites, especially considering the accurate measurement of CTOA.

Figure 12a shows the measured force as function of the applied cross-head displacement ($u_{22}$) for PS specimens subjected to mode I loading at fiber orientations of $0^{\circ }$ and $90^{\circ }$. Figure 12b illustrates the measured force as function of the estimated crack growth (da) using the proposed approach. For both fiber orientations, a similar behaviour can be noticed and a satisfactory repeatability was observed. It is clear that the fibers oriented at $0^{\circ }$ are most effective against crack propagation [34]. As a final remark, although there are some filaments between the unidirectional fibers, the influence of these filaments on mode I loading can be neglected [35].

## 4. Conclusions

This work addressed an alternative DIC-based experimental approach to estimate the crack tip position and evaluate important fracture parameters (CTOD, CTOA and crack growth) of a soft composite reinforced with unidirectional warp knitted cotton fabric subjected to mode I fracture using a pure shear configuration. DIC-based techniques have been used to extract the fracture kinematics of specimens under mode I loading. However, in the presence of extensible fibers, this image-based technique fails. The stretched fibers can be considered a random pattern that compromises the DIC measurements near the crack tip, introducing errors. To overcome this limitation, DIC was associated with image segmentation in the literature to isolate the crack boundaries of homogeneous and nonhomogeneous materials. Nevertheless, even image segmentation can not remove the stretched fibers of the considered soft composite and, at a certain level, can induce a loss of crack boundaries and underestimate the crack tip position. In addition, some algorithm-based methodologies are complicated to be reproduced, and the associated computational cost can be expensive. The crack tip position was fast and accurately detected using the proposed methodology, overcoming the problem associated with the random fiber distribution. The crack propagation was measured for both sharp and blunt crack opening shapes. As a final remark, the proposed methodology can be extended for mixed mode loading condition and will be addressed in future works. Moreover, different correction functions can be proposed for different materials in view of improving this methodology. This study can be extended to biological fibrous soft tissues since the synthetic composite used in this work presents a similar behaviour, e.g., both materials can be modelled, at given assumptions, as transversely isotropic hyperelastic materials.

## Figures and Tables

**Figure 1 materials-15-02413-f001:**
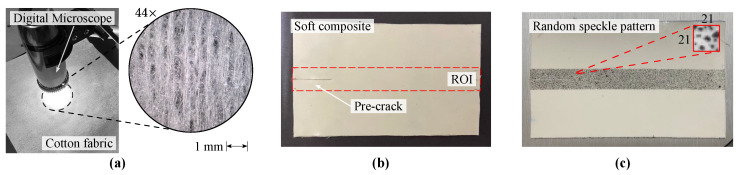
(**a**) Structure of the cotton knit fabric, (**b**) manufactured soft composite with depicted Region of Interest (ROI) and (**c**) detail of the random speckle pattern and subset size of 21 × 21 pixels^2^.

**Figure 2 materials-15-02413-f002:**
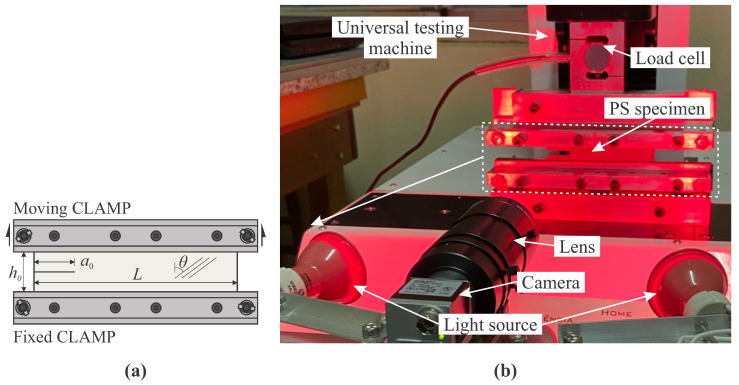
(**a**) Schematic representation of the PS specimen with a pre-crack of $a_{0}=20$ mm and (**b**) experimental set-up.

**Figure 3 materials-15-02413-f003:**
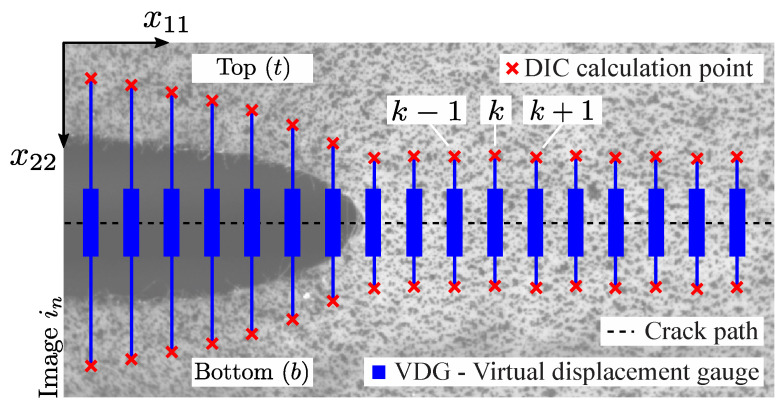
DIC-based virtual displacement gauge (VDG).

**Figure 4 materials-15-02413-f004:**
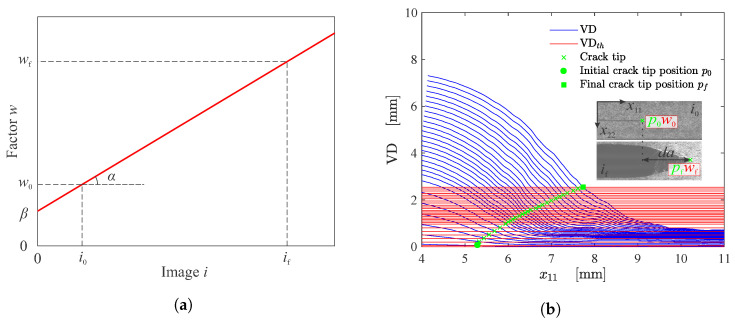
(**a**) Linear threshold correction factor and (**b**) estimation of crack tip position (*p*) from VDG approach. Blue curves depict the VD curve for each instant *n* and the red lines indicate the adjusted threshold line-cut. The intersection between ${\mathrm{VD}}_{th}\left(i_{n}\right)$ and $\mathrm{VD}(k,i_{n})$ corresponds to the crack tip position at instant *n*.

**Figure 5 materials-15-02413-f005:**
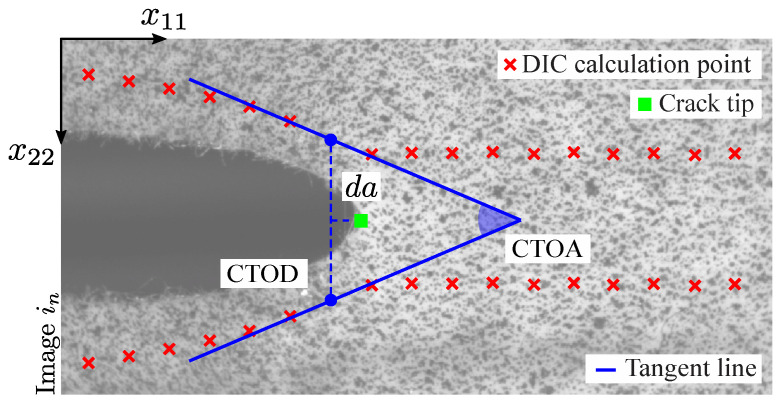
CTOD and CTOA estimation.

**Figure 6 materials-15-02413-f006:**
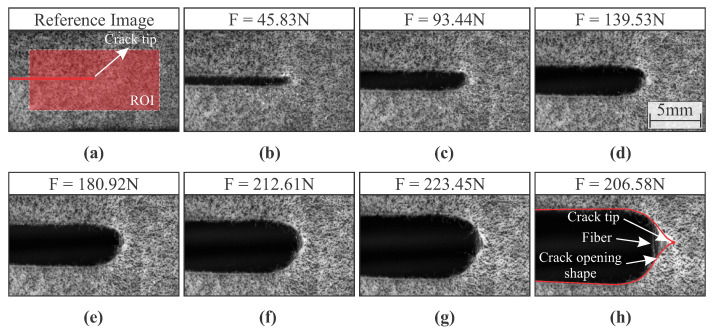
Crack growth in a fibrous hyperelastic soft composite with fibers oriented at $0^{\circ }$: (**a**) undeformed state with depicted Region of Interest (ROI); (**b**) deformed states with applied loads of 45.83 N; (**c**) 93.44 N; (**d**) 139.53 N; (**e**) 180.92 N; (**f**) 212.61 N; (**g**) 223.45 N; (**h**) 206.58 N.

**Figure 7 materials-15-02413-f007:**
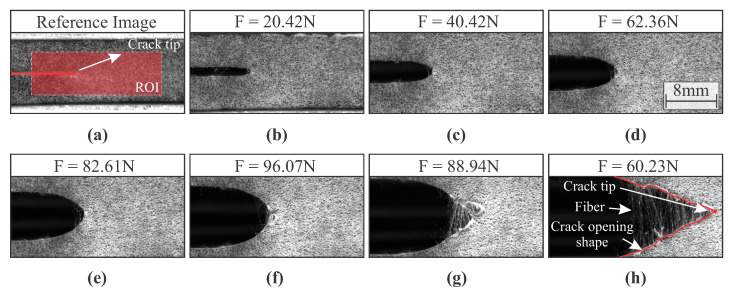
Crack growth in a fibrous hyperelastic soft composite with fibers oriented at $90^{\circ }$: (**a**) undeformed state with depicted Region of Interest (ROI); (**b**) deformed states with applied loads of 20.42 N; (**c**) 40.42 N; (**d**) 62.36 N; (**e**) 82.61 N; (**f**) 96.07 N; (**g**) 88.94 N; (**h**) 60.23 N.

**Figure 8 materials-15-02413-f008:**
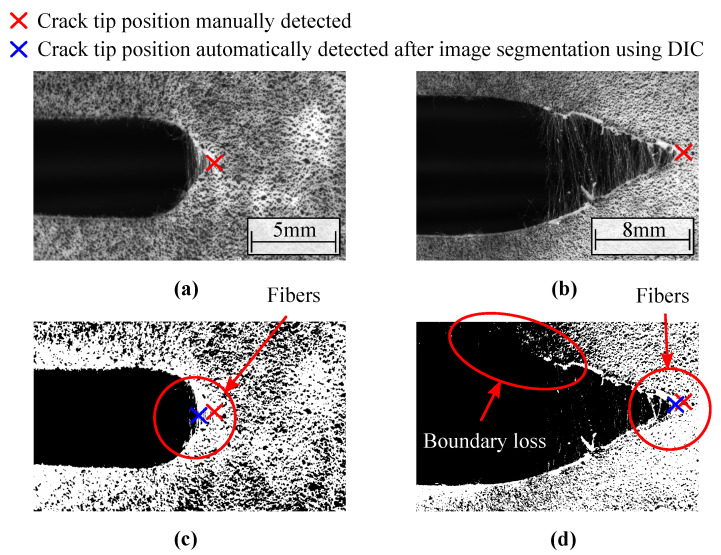
Image segmentation problematic. Captured image of soft composite with fibers oriented at (**a**) $0^{\circ }$ and (**b**) $90^{\circ }$, and image segmentation considering (**c**) $\theta =0^{\circ }$ and (**d**) $\theta =90^{\circ }$. The red cross symbol depicts the manually detected crack tip position while the blue cross depicts the automatically detected crack tip position using DIC after image segmentation.

**Figure 9 materials-15-02413-f009:**
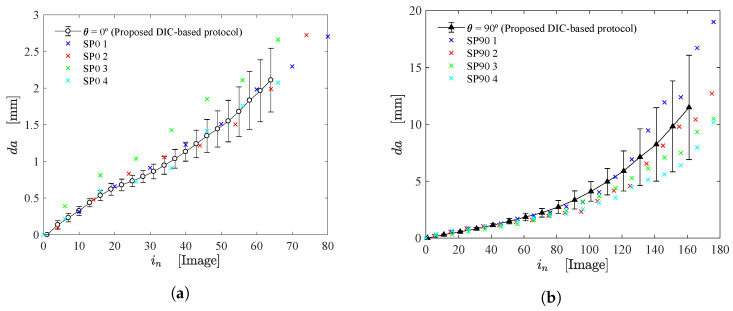
Crack growth (da) as function of the number of captured images for fiber orientations of $0^{\circ }$ (**a**) and $90^{\circ }$ (**b**). The continuous curve depicts the proposed method using DIC while the cross symbols show the results of manual detection.

**Figure 10 materials-15-02413-f010:**
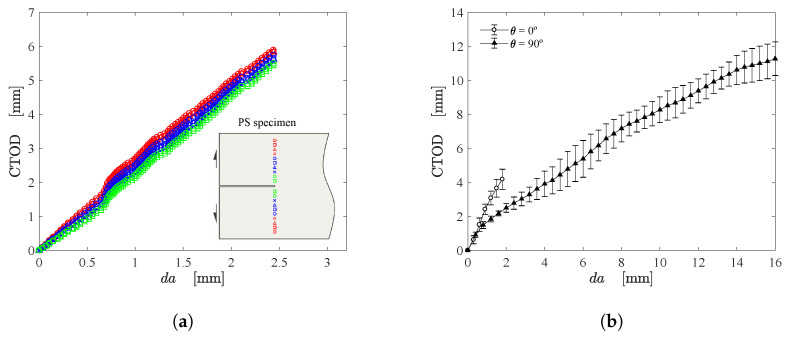
(**a**) CTOD estimation using different DIC calculation points and (**b**) CTOD-da curve for fiber orientations of $0^{\circ }$ and $90^{\circ }$.

**Figure 11 materials-15-02413-f011:**
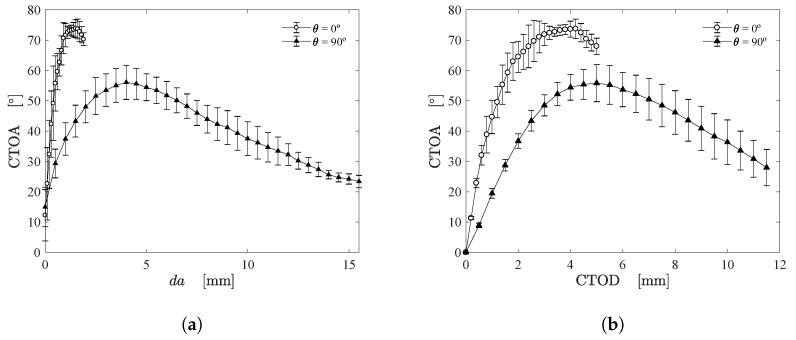
(**a**) CTOA as function of crack growth and (**b**) CTOD.

**Figure 12 materials-15-02413-f012:**
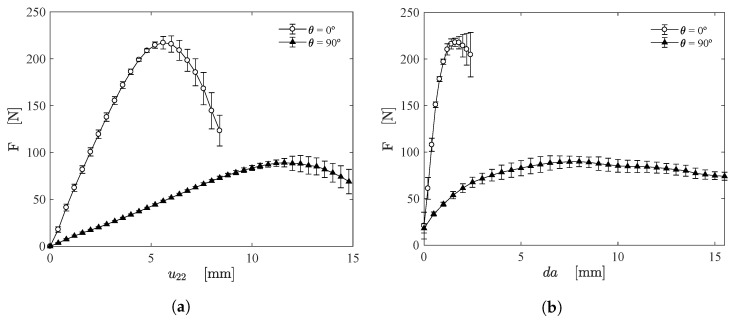
(**a**) Force as function of the cross-head displacement ($u_{22}$) and (**b**) the crack growth (da).

**Table 1 materials-15-02413-t001:** *iCorrVision-2D* DIC settings.

Correlation Parameters	Value
Reference subset size	21 × 21 pixels${}^{2}$
Target subset size	71 × 71 pixels${}^{2}$
Mean calibration factor	38.7 pixels/mm
Elements in $x_{11}$ direction	200 (for $\theta =0^{\circ }$) and 400 (for $\theta =90^{\circ }$)
Elements in $x_{22}$ direction	1
Image interpolation (Bicubic spline)	10×
Subpixel level (Bicubic spline)	10
Correlation function	TM_CCORR_NORMED [33]
Correlation criterion	0.9
Correlation approach	Incremental

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
