# Peer review of "An Alternative Digital Image Correlation-Based Experimental Approach to Estimate Fracture Parameters in Fibrous Soft Materials"

_materials, 2022, doi:10.3390/ma15072413_

Round 1

Reviewer 1 Report

Dear Authors,
I consider your manuscript as valuable especially due to the significant contribution to the development of the methodology of non-contact measurements of materials deformation during mechanical testing.
Below please find several remarks:
- to Introduction section, as well as to discussion, where you analyze the presence of filaments (lines 206-208), please, consider adding the following citation, where wood-based composite with regenerated cellulose fibers has been tested, and a similar effect (bridging) has been found: Kowaluk G. (2014): Properties of lignocellulosic composites containing regenerated cellulose fibers. BioResources 9(3), 5339-5348
- line 91 - you said that "The camera-lens set was positioned parallel to the plane surface of the PS specimen"; I think the camera-lens axis was perpendicular to the plane surface of the specimen
- line 94: please convert "50 Kgf" to N

Best regards!

Author Response

Journal: Materials (ISSN 1996-1944)

Manuscript ID: materials-1656316

Type: Article

Title: An alternative DIC-based experimental approach to estimate fracture parameters in fibrous soft materials

Authors: João Filho*, José Xavier*, Luiz Nunes

The authors would like to thank the reviewers for the valuable comments provided to the original version of the manuscript. Accordingly, all questions have been addressed in view to improve the manuscript. A revised version of the paper was prepared, in which carefully considerations of all queries mentioned in the reviewers' comments were addressed. Changes are highlighted in blue colour in the new version of the manuscript. Please, find enclosed a point-by-point response to the reviewer’s comments.

Reviewer #1

Dear Authors,

I consider your manuscript as valuable especially due to the significant contribution to the development of the methodology of non-contact measurements of materials deformation during mechanical testing.

Authors: The authors would like to thank you for the general positive remark on the contribution of our paper.

Below please find several remarks:

- to Introduction section, as well as to discussion, where you analyze the presence of filaments (lines 206-208), please, consider adding the following citation, where wood-based composite with regenerated cellulose fibers has been tested, and a similar effect (bridging) has been found: Kowaluk G. (2014): Properties of lignocellulosic composites containing regenerated cellulose fibers. BioResources 9(3), 5339-5348

Authors: The suggested citation has been included in the revised manuscript when discussing the fiber orientations with regard do bridging effects in mode I loading.

- line 91 - you said that "The camera-lens set was positioned parallel to the plane surface of the PS specimen"; I think the camera-lens axis was perpendicular to the plane surface of the specimen

Authors: This text has been modified and improved in the revised manuscript.

- line 94: please convert "50 Kgf" to N

Authors: In the revised manuscript the value of 500 N has been updated.

Reviewer 2 Report

The article titled “An alternative DIC-based experimental approach to estimate fracture parameters in fibrous soft materials” uses DIC to predict the crack initiation and propagation in fibrous composites. The approach and the methodology is unique and of great importance in the field of fracture mechanics. I recommend that the minor suggestions here must be addressed before the manuscript can be considered for publication.

  1. Include the full form of DIC in the title
  2. Change keyword to Mode I fracture or Mode I loading
  3. Literature on DIC based measurements must be explained briefly in the Introduction. For instance, past studies on DIC based measurements and their respective findings, advantages and drawbacks, if any.
  4. Line 64. Which biological tissue?
  5. Indicate the full form of ROI in the figure 5 and 6 captions
  6. Include scale bar in Figure 7.
  7. From Figure 8, it can be seen that DIC based measurements become less accurate or has a greater standard deviation with the increase in the number of images detected. This is true in both fiber orientation cases. Explain the reason.

Author Response

Journal: Materials (ISSN 1996-1944)

Manuscript ID: materials-1656316

Type: Article

Title: An alternative DIC-based experimental approach to estimate fracture parameters in fibrous soft materials

Authors: João Filho*, José Xavier*, Luiz Nunes

The authors would like to thank the reviewers for the valuable comments provided to the original version of the manuscript. Accordingly, all questions have been addressed in view to improve the manuscript. A revised version of the paper was prepared, in which carefully considerations of all queries mentioned in the reviewers' comments were addressed. Changes are highlighted in blue colour in the new version of the manuscript. Please, find enclosed a point-by-point response to the reviewer’s comments.

Reviewer #2

The article titled “An alternative DIC-based experimental approach to estimate fracture parameters in fibrous soft materials” uses DIC to predict the crack initiation and propagation in fibrous composites. The approach and the methodology is unique and of great importance in the field of fracture mechanics. I recommend that the minor suggestions here must be addressed before the manuscript can be considered for publication.

  1. Include the full form of DIC in the title

Authors: The title of the paper has been changed, as suggested by the reviewer.

  1. Change keyword to Mode I fracture or Mode I loading

Authors: The new keyword has been changed to Mode I fracture.

  1. Literature on DIC based measurements must be explained briefly in the Introduction. For instance, past studies on DIC based measurements and their respective findings, advantages and drawbacks, if any.

Authors: The Introduction section has been improved by extending the literature review about DIC based measurements in the framework of fracture mechanics.

  1. Line 64. Which biological tissue?

Authors: The biological tissue has been specified in the revised manuscript.

  1. Indicate the full form of ROI in the figure 5 and 6 captions

Authors: Region of Interest (ROI) has been included in captions of both Figures 6 and 7 in the revised manuscript (Figures 5 and 6, respectively, in the submitted manuscript).

  1. Include scale bar in Figure 7.

Authors: A scale bar has been included in Figure 8 (Figure 7 in the submitted manuscript).

  1. From Figure 8, it can be seen that DIC based measurements become less accurate or has a greater standard deviation with the increase in the number of images detected. This is true in both fiber orientation cases. Explain the reason.

Authors: The standard deviation is expected to be higher with the increase of the crack growth (da). In fact, the standard deviation increases proportionally with the increase of da. Therefore, taking the normalized standard deviation in terms of da, they are approximately the same with the increase in the number of images for both fiber orientations. It should be highlighted that the standard deviation of the proposed methodology followed the data dispersion of the manually detected crack tip, according to Figure 8 (now, Figure 9 in the revised manuscript). The standard deviations are high due to the complexity of the material tested in this work (fabrication process and random distribution of fibers).

Reviewer 3 Report

The article is devoted to studying of new methodology for crack tip detection of soft fiber-reinforced composites. The results of the work are of interest to readers in the field of fiber composite materials, but the article requires a minor revision:

1) Decrease excessive self-citation in Introduction. There are 3 self-citations in lines 42-43: “(DIC) technique has been used to measure the crack propagation from full-field displacements with sub-pixel resolution [13 – 16].” (citations 14-16 are self-citations, one of them should be enough) and 3 self-citations in lines 48-50 “Besides, the crack tip opening  displacement has been computed from DIC measurements to directly determine cohesive laws of biological tissues [19–21].” (citations 19-21 are self-citations, one of them should be enough).

2) Lines 51-52: “For nonhomogeneous materials, the presence of fibers at the crack tip can lead to a challenging crack tip detection.” Please, specify if materials are nonhomogeneous due to the presence of fibers or even without them?

3) In line 67 is stated that samples were “made of unidirectional fibers” but it is hard to distinguish any fibers in figures 2, 4, 5-7. Please provide photographs showing the structure of the composite with fibers for a visual understanding of the orientation and density of fiber filling.

4) The paper does not reflect the change in what characteristics of the samples is easily and correctly taken into account in the method? Among such characteristics may be the geometric dimensions of the samples (in particular, the thickness), the concentration of fibers, the dimensions of the fibers, and others.

5) Add comparative results in a separate figure or graph about what results are obtained when using “The commonly used image-based techniques”.

Author Response

Journal: Materials (ISSN 1996-1944)

Manuscript ID: materials-1656316

Type: Article

Title: An alternative DIC-based experimental approach to estimate fracture parameters in fibrous soft materials

Authors: João Filho*, José Xavier*, Luiz Nunes

The authors would like to thank the reviewers for the valuable comments provided to the original version of the manuscript. Accordingly, all questions have been addressed in view to improve the manuscript. A revised version of the paper was prepared, in which carefully considerations of all queries mentioned in the reviewers' comments were addressed. Changes are highlighted in blue colour in the new version of the manuscript. Please, find enclosed a point-by-point response to the reviewer’s comments.

Reviewer #3

The article is devoted to studying of new methodology for crack tip detection of soft fiber-reinforced composites. The results of the work are of interest to readers in the field of fiber composite materials, but the article requires a minor revision:

1) Decrease excessive self-citation in Introduction. There are 3 self-citations in lines 42-43: “(DIC) technique has been used to measure the crack propagation from full-field displacements with sub-pixel resolution [13 – 16].” (citations 14-16 are self-citations, one of them should be enough) and 3 self-citations in lines 48-50 “Besides, the crack tip opening displacement has been computed from DIC measurements to directly determine cohesive laws of biological tissues [19–21].” (citations 19-21 are self-citations, one of them should be enough).

Authors: Accordingly, self-citations have been reduced to only one citation in the revised manuscript.

2) Lines 51-52: “For nonhomogeneous materials, the presence of fibers at the crack tip can lead to a challenging crack tip detection.” Please, specify if materials are nonhomogeneous due to the presence of fibers or even without them?

Authors: The matrix was made of silicone rubber, which is a homogeneous rubber-like material. Therefore, the fiber-reinforced soft material is nonhomogeneous due to the presence of fibers. The text has been adjusted in the revised manuscript.

3) In line 67 is stated that samples were “made of unidirectional fibers” but it is hard to distinguish any fibers in figures 2, 4, 5-7. Please provide photographs showing the structure of the composite with fibers for a visual understanding of the orientation and density of fiber filling.

Authors: An additional figure has been included in the revised manuscript to illustrate the fiber arrangement. Please see the Materials and Methods section.

4) The paper does not reflect the change in what characteristics of the samples is easily and correctly taken into account in the method? Among such characteristics may be the geometric dimensions of the samples (in particular, the thickness), the concentration of fibers, the dimensions of the fibers, and others.

Authors: The aim of this paper was not to investigate the different configurations of the soft composite samples, but the alternative DIC-based methodology to easily estimate the fracture kinematics, such as crack growth, CTOD and CTOA. However, this issue should be addressed in future works to investigate the influence of these material characteristics on the fracture parameters using the proposed methodology.

5) Add comparative results in a separate figure or graph about what results are obtained when using “The commonly used image-based techniques”.

Authors: In the submitted manuscript, Figure 7 (now, Figure 8 in the revised manuscript) illustrates the underestimation of the crack tip when employing image segmentation before carrying out the Digital Image Correlation (blue cross symbols). This image-based technique is commonly used in several works to detect the crack tip and should be avoided in the presence of fiber-reinforced soft materials, as pointed out in the manuscript. The red cross symbols illustrate the manually detected crack tip position. A comparative result is depicted in Figure 8 (now, Figure 9 in the revised manuscript). The scattered data (cross symbols) were obtained using image-based manual detection and the curves were generated by the proposed DIC-based methodology using a linear correction function.

Reviewer 4 Report

This paper deals with the crack tip detection of fibrous soft composites based on the Digital Image Correlation (DIC) method. The crack growth (da), Crack Tip Opening Displacement (CTOD) and Crack Tip Opening Angle (CTOA) were determined using pure shear specimens under mode I fracture. Overall, the paper is well organized and written. The following comments can further improve the quality of the paper.

* Introduction

#1 DIC method was powerful to obtain full-field strain of tested specimens, which was widely used in the study of fracture mechanics of composites. The authors should add more application instances of DIC in the research background. The references of “Effect of fiber hybridization types on the mechanical properties of carbon/glass fiber reinforced polymer composite rod (Mechanics of Advanced Materials and Structures, 2021)” and “Combined effects of sustained bending loading, water immersion and fiber hybrid mode on the mechanical properties of carbon/glass fiber reinforced polymer composite (Composites Structure)” may provide more support information for the present introduction. It is suggested that the authors summarize the related analysis on full-field strain by DIC.

* Materials and Methods

#2 The detailed manufacture method and process of soft composite should be given, and provide the figure of soft composite.

#3 Is there a quality control requirement (such as size and number) for speckle? How to control the speckle quality? This is very important to obtain a uniform and stable full-field strain distribution.

#4 In Figure 8, please explain why the crack growth and the error bar of q=90o was larger than that of q=0o?

#5 Please explain the meaning of the inflection point of curves in Figure 10.

#6 DIC method was widely used for mode I fracture, please highlight the innovation and contribution (method or theory) of present paper in the conclusions.

Author Response

Journal: Materials (ISSN 1996-1944)

Manuscript ID: materials-1656316

Type: Article

Title: An alternative DIC-based experimental approach to estimate fracture parameters in fibrous soft materials

Authors: João Filho*, José Xavier*, Luiz Nunes

The authors would like to thank the reviewers for the valuable comments provided to the original version of the manuscript. Accordingly, all questions have been addressed in view to improve the manuscript. A revised version of the paper was prepared, in which carefully considerations of all queries mentioned in the reviewers' comments were addressed. Changes are highlighted in blue colour in the new version of the manuscript. Please, find enclosed a point-by-point response to the reviewer’s comments.

Reviewer #4

This paper deals with the crack tip detection of fibrous soft composites based on the Digital Image Correlation (DIC) method. The crack growth (da), Crack Tip Opening Displacement (CTOD) and Crack Tip Opening Angle (CTOA) were determined using pure shear specimens under mode I fracture. Overall, the paper is well organized and written. The following comments can further improve the quality of the paper.

* Introduction

#1 DIC method was powerful to obtain full-field strain of tested specimens, which was widely used in the study of fracture mechanics of composites. The authors should add more application instances of DIC in the research background. The references of “Effect of fiber hybridization types on the mechanical properties of carbon/glass fiber reinforced polymer composite rod (Mechanics of Advanced Materials and Structures, 2021)” and “Combined effects of sustained bending loading, water immersion and fiber hybrid mode on the mechanical properties of carbon/glass fiber reinforced polymer composite (Composites Structure)” may provide more support information for the present introduction. It is suggested that the authors summarize the related analysis on full-field strain by DIC.

Authors: The Introduction section has been improved by extending the literature review about DIC measurements in the framework of fracture mechanics. The proposed methodology is a post-processing algorithm based on displacement fields, rather than strain fields.

* Materials and Methods

#2 The detailed manufacture method and process of soft composite should be given, and provide the figure of soft composite.

Authors: The detailed manufacture method and process of the fiber-reinforced soft material have been included in the revised manuscript. Additionally, Figure 1 has been added to illustrate the fiber arrangement and the soft composite.

#3 Is there a quality control requirement (such as size and number) for speckle? How to control the speckle quality? This is very important to obtain a uniform and stable full-field strain distribution.

Authors: According to the International Digital Image Correlation Society (iDICs), a suitable average feature size should be between 3 and 5 pixels, the subset must contain approximately 50% of speckle density and at least 3 features to guarantee a good match between images. Moreover, it was recommended by the iDICs that a subset size of 21 x 21 pixels2 can be a more suitable minimum subset size for DIC measurements. The correlation procedure addressed in this work followed the iDICs guidelines that can be encountered in “International Digital Image Correlation Society, Jones, E.M.C. and Iadicola, M.A. (Eds.) (2018). A Good Practices Guide for Digital Image Correlation”. The average speckle size of 4.9 pixels was obtained in this work and a black/white ratio of 1.04 was encountered. No decorrelation was seen for the considered DIC settings shown in Table 1 and a good displacement distribution was observed. Please find in the revised manuscript the quality control data and the detailed random speckle pattern (Figure 1(c)) in the Materials and methods section.

#4 In Figure 8, please explain why the crack growth and the error bar of q=90o was larger than that of q=0o?

Authors: The standard deviation is expected to be higher with the increase of the crack growth (da). In fact, the standard deviation increases proportionally with the increase of da. Therefore, taking the normalized standard deviation in terms of da, they are approximately the same with the increase in the number of images for both fiber orientations. It should be highlighted that the standard deviation of the proposed methodology followed the data dispersion of the manually detected crack tip, according to Figure 8 (now, Figure 9 in the revised manuscript). The standard deviations are high due to the complexity of the material tested in this work (fabrication process and random distribution of fibers). Additionally, since the specimens with fibers oriented at the loading direction (theta = 0º) are stiffer than ones with fibers at theta = 90º (see Figure 12 of the revised manuscript), the crack growth is lower for samples with fibers oriented at theta = 0º, as expected.

#5 Please explain the meaning of the inflection point of curves in Figure 10.

Authors: In the submitted manuscript, Figure 10 (now, Figure 11 in the revised manuscript) does not have any inflection point. The authors assume that the reviewer was referring to the maximum point. As pointed out in the Results and discussion section of the submitted manuscript, the crack growth undergoes a change from blunt to a typical sharp crack for both sample configurations. The maximum point can be explained by the shape change in the crack propagation. In fact, after a large crack growth, the shape change induces a decrease of the CTOA, as illustrated in Figures 6 and 7 of the revised manuscript.

#6 DIC method was widely used for mode I fracture, please highlight the innovation and contribution (method or theory) of present paper in the conclusions.

Authors: The innovation and contributions have been highlighted in the revised manuscript.

Round 2

Reviewer 4 Report

It can be accepted in the present form.